# Correlates of Health-Related Quality of Life in Community-Dwelling Older Adults in Guadeloupe (French West Indies): Results from the KASADS Study

**DOI:** 10.3390/ijerph20043004

**Published:** 2023-02-09

**Authors:** Nadine Simo, Denis Boucaud-Maitre, Pierre Gebhard, Roxane Villeneuve, Leila Rinaldo, Jean-François Dartigues, Moustapha Drame, Maturin Tabue-Teguo

**Affiliations:** 1CHU de Martinique, 97261 Fort-de-France, France; 2Equipe EpiCliV, Université des Antilles, 97157 Pointe-à-Pitre, France; 3Centre Hospitalier le Vinatier, 69500 Bron, France; 4CHU de Guadeloupe, 97110 Pointe-à-Pitre, France; 5Equipe ACTIVE, ISERM 1219, Univ. Bordeaux, 33000 Bordeaux, France

**Keywords:** correlates, HRQoL, aged, Guadeloupe

## Abstract

Objectives: The aim of this study was to determine the correlates of health-related quality of life (HRQoL) in community-dwelling older adults in Guadeloupe. Methods: We used the Karukera Study of Aging-Drugs Storage (KASADS), an observational, cross-sectional study on community-dwelling older people living in Guadeloupe. A visual analogue scale ranging from 0 to 100 was used to assess HRQoL. Results: The study sample consisted of 115 patients aged 65 years or older; 67.8% were women. Participants were 76 (±7.8) years old with a mean HRQoL of 66.2 (±20.3). The correlates of HRQoL were complaints of pain (*p* < 0.001) and IADL dependency (*p* = 0.030) after adjustment. We found no significant interactions between HRQoL and other variables such as marital status, socio-educational level and cognitive decline. Conclusions: Pain and IADL dependency were independently associated with lower HRQoL in community-dwelling older people in Guadeloupe.

## 1. Introduction

Quality of life (QoL) refers to “an individual’s perception of their position in life in the context of the culture and value systems in which they live and in relation to their goals, expectations, standards and concerns” [1].

It is a multidimensional concept that reflects an individual’s subjective attitude and experience of physical, mental and social health [2]. 

The increasing process of demographic aging of the population brings with it new tasks for state and local administrations, which will require the formulation of objectives for a policy that is conducive to meeting the specific needs of older people. A set of these needs characterizes the quality of life of older people, which in everyday life is reflected in their health.

Two main types of factors determine the quality of life of an individual: objective factors (standard of living, health status, functional abilities, social participation and contacts, leisure activities) and subjective factors (subjective health, the meaning of life) [2,3,4]. Some symptoms such as pain are referenced as correlates of quality of life [4]. A proxy with proper training can assess objective QoL. Subjective QoL reflects someone’s feelings about their psychological, social and medical situation according to their standards and comparisons with others [5]. It is a multidimensional measure representing a person’s acceptance of their condition, taking into account several dimensions of health (physical, mental and the support of those around them). Therefore, older adults’ “objective” state of health would only partially determine their subjective health status, a standard proxy for health-related quality of life (HRQoL) which considers health’s impact on everyday life. Indeed, this satisfaction is primarily the result of the comparison they make with the state of health of other seniors their age [6].

One study defines health-related quality of life (HRQoL) as the value assigned to lifespan as modified by impairment, functional states, perception and social opportunities that are influenced by illness, injury, treatment or policy [7].

These social comparisons also help define expectations or aspirations when assessing QoL. They partly explain the “well-being paradox” (the fact that older people with significant limitations in their daily lives may rate their well-being positively) [8]. HRQoL may also be useful indirectly to measure the impact of a disease or assess the effectiveness of a treatment or care plan without a purely medical perspective. Many factors are associated with lower levels of HRQoL. Among them, pain shows the strongest association, whether people suffer from cognitive impairment or not. Severity and insight seem to modulate the association with HRQoL in people with cognitive impairment: older people with mild or moderate dementia and good insight present lower levels of quality of life than people without cognitive impairment and people with moderate/severe dementia with low insight.

Health-related quality of life (HRQoL) consistently and significantly predicts adverse health outcomes (including disability, neurodegenerative diseases and mortality) [9,10].

Previous research has demonstrated that frailty is associated with lower HRQoL in oncology cohorts, and patients have reported that HRQoL is increasingly considered a significant outcome measure for cancer treatment [11,12].

Despite a growing body of studies on older people’s HRQoL, few focus on older people’s HRQoL in the Caribbean. Girvan describes the Caribbean region as an “ethnohistoric zone and a transnational community”, highlighting the cultural similarities between the territories. Demographic trends show a steady decline in birth and mortality rates, coupled with a distinctive international migration pattern that inflates the rate of older people [13]. For instance, Guadeloupe, a French-Caribbean territory, has been the youngest French territory for decades, but it should become one of the oldest in 10 years. Compared to mainland France, the older Caribbean population suffers from a higher prevalence of dementia, specific Parkinsonian syndromes probably linked to environmental factors, metabolic syndromes and, overall, higher dependency rates at a younger age [14]. The aim of this study was to determine the correlates of health-related quality of life (HRQoL) in community-dwelling older adults in Guadeloupe. Our study took advantage of the KASADS study database, a baseline screening conducted in Guadeloupe.

## 2. Methods

### 2.1. Study Design

This observational, cross-sectional study in Guadeloupe (a French department in the Caribbean) uses the Karukera Study Aging Drug Storage (KASADS) study [15]. In summary, the KASADS study aimed to assess the association between drug storage and the risk of frailty in older people (>65 years) living at home in Guadeloupe. In the KASADS cohort, two general medicine interns trained in geriatric assessment tools collected demographic data regarding frailty syndrome, medication storage, cognitive function, algal complaints, functional status and health-related quality of life. To limit selection bias, the participating general practitioners systematically asked all their patients who met the inclusion criteria to participate in the study until they each reached 10 participants. The patients who agreed to participate received an information leaflet specifying the modalities of the study and provided written informed consent. The Ethics Committee of the Guadeloupe University Hospital approved this study.

### 2.2. Variables

Subjective HRQoL was collected using a visual analogue scale (VAS) assessing subjective health, a proxy for HRQoL. This continuous scale is a 100 mm line with the ends marked “very poor” and “very good”. Participants were asked to draw a line perpendicular to the VAS line to represent their perceived health-related quality of life. The score was the distance in centimeters from the lower end to the mark [16].

### 2.3. Other Variables

We used the Instrumental Activities of Daily Living scale (IADL) [17] and Katz’ ADL scale [18] to assess the functional status of the participants. The Lawton’s IADL scale measures four items: using the telephone, shopping, preparing food, housekeeping, doing laundry, using transportation, handling medications and handling finances. The score is from 0 to 4; a score of 0 indicates total autonomy and 4 indicates total dependency. The Katz’ ADL scale is a six-item scale assessing independence in six basic ADL: bathing, toileting, transferring, eating, dressing and incontinence. For each activity, a score of 1 indicates complete autonomy, a score of 0.5 indicates partial autonomy and a score of 0 indicates complete dependency. The Mini-Mental State Examination (MMSE) was used to assess cognitive functions. This 30-item scale assesses the severity of cognitive deterioration through items of orientation, learning, attention and arithmetic, memory, language and constructive praxis. The scale is rated from 0 to 30, reflecting the level of cognitive impairment. The threshold for suspicion of cognitive impairment was 24/30 [19]. The participants estimated their pain with a visual analogue scale. The participants’ socio-demographic characteristics and comorbidities were also collected: age, sex, educational level and marital status for sociodemographic variables; absence or presence of diabetes, hypertension, dyslipidemia and Body Mass Index (BMI) for comorbidities.

### 2.4. Statistical Analysis

Quantitative variables were expressed as mean ± standard deviation, and qualitative variables were expressed as percentages. Student’s *t*-test was used to assess the significance of associations between categorical and continuous variables, and Pearson’s correlation was used to assess the correlation between continuous variables. Variables at a 20%-threshold in the univariate analysis were considered in a multivariable analysis using a linear regression. At last, an interaction term between pain and cognitive impairment on quality of life was searched in a multivariable analysis using a linear regression model. Missing data were not imputed. We used a fixed *p*-value cut-off of 0.05 to determine significance. All analyses were performed with the RStudio software (v.3.0.2. 21).

## 3. Results

### Study Sample

We included 115 community-dwelling people aged 65 and older. The average age of the participants was 76.0 ± 7.8 years, with a BMI of 26.8 ± 5.3, and 67.8% were women. 43.5% had diabetes, 87.0% had hypertension, 45.2% had dyslipidemia and 58.3% had obesity. The mean HRQoL score was 66.2 ± 20.3 and 51.6 ± 21.7 for pain. The mean IADL score was 3.4 ± 1.0, and the mean ADL score was 5.79 (±0.83) (see Table 1).

Table 2 shows that there is an association between HRQoL and pain (correlation coefficient: 0.298; *p* = 0.001), between HRQoL and IADL (coefficient: 0.254; *p* = 0.006) and between HRQoL and a low socio-educational level (*p* = 0.014).

Table 3 shows the correlates of HRQoL in multivariable analysis. The adjustment variables retained for the multivariable analysis were those exhibiting a significant association (*p* < 0.20) with HRQoL in univariate analysis. Pain (*p* < 0.001) and IADL (*p* = 0.030) were significantly associated with HRQoL. There was no significant association between cognitive impairment, socio-educational level and quality of life, and there was no interaction effect between cognitive impairment and pain on quality of life (*p* = 0.741).

Associations between HRQoL and categorical variables.

## 4. Discussion

Our objective was to determine the correlates of HRQoL in the general older population in Guadeloupe. We found that complaints of pain were associated with poorer HRQoL. Several studies have also found that pain [20,21,22] was associated with an altered HRQoL. In 2003, Jakobsson [23,24] found this association in the Swedish population over 70—including after stratification on age. Lacey et al. [22] found similar results in Ireland in 2014 and established a relationship between pain and physical and mental components of HRQoL, using the SF-12 questionnaire, a validated HRQoL assessment. It is difficult to report on the pain experience itself. There are several components to pain, including the affective-emotional one. Chronic pain is strongly associated with incident anxiety and depression [20,21], which contributes to the deterioration of quality of life.

The diagnosis of pain in older people is challenging. People with cognitive disorders may underestimate their pain in self-rated scales due to difficulties in perception, expression, analytical faculties or global understanding [25,26]. Hunt et al. [27] showed that proxies tend to report more pain than participants. They found that up to 30% of older people suffering from various types of pain took no pain medication [27].

As untreated pain is the cause of functional limitations and impacts quality of life, pain management is essential. Assessing physical quality of life requires self- or proxy-assessment, which accounts for context and comorbidities, to select the appropriate treatment. Although this association is often found in the literature, alleviating pain and complaints of pain does not necessarily improve quality of life.

In our study, impairment of IADL was associated with a lower quality of life. This result is even more relevant as studies show an association between pain and IADL impairment. Covinsky et al. in 2009 and Shega et al. in 2010 [28,29] showed that significant pain correlates with functional limitations and their early appearance, especially with high-prevalence pathologies in later life, such as osteoarticular disorders. The location of the pain (for example, lower limbs) is a risk factor for dependence and lower quality of life.

We found no association between neurocognitive impairment and quality of life (correlation coefficient: 0.129; *p* = 0.171). Although the data available in the literature generally agree with this result, it depends on the population or the assessment instruments. In a study on subjective quality of life, Baptista et al. show that people with mild cognitive impairment were more aware of their impairment than those with severe cognitive impairment. Awareness of the disorder was associated with lower reported quality of life than those unaware [30]. Hill et al. also showed that the mere complaint of subjective cognitive impairment would negatively impact the quality of life of individuals compared to those without such complaints [31]. On the other hand, in advanced stages of neurocognitive disorders, certain studies such as that of Selwood et al. or Missotten et al. show no association between the evolution of cognitive disorders and quality of life over time in cohort studies [32,33].

Assessing subjective HRQoL is a challenge in older people suffering from cognitive impairment. Self-report methods may be inappropriate in people with severe disorders, as they involve understanding the complex concept of quality of life. Nevertheless, various validated HRQoL scales exist for patients with dementia, depending on the type of dementia, the degree of severity of the disease and the place of living of the patient [34].

In our study, in the absence of other assessments of cognition, only the threshold of 24/30 on the MMSE allowed us to classify our participants according to their level of cognitive impairment. A more thorough diagnostic approach could have allowed us to characterize them more precisely and propose complementary, more appropriate assessments.

Other studies also demonstrate the significant contribution of social support to predict quality of life [35] and serve as a buffer to stress [36].

The “quality of life gap theory” defines quality of life as the gap between someone’s ideal life and the perceived reality. Individuals with poor “objective” health might thus report good quality of life when reality matches their expectations [37,38].

Other factors are crucial to the quality of life of older people, for instance, social support from family or professional caregivers or participation in social activities (seniors’ club, Alzheimer’s network). This type of assistance reduces the daily organizational burden and the preservation of the social fabric and is a pivotal determinant of quality of life [35]. Some variables that were not significant in multivariable analysis were not commented on, such as the lack of a diploma and living alone. The inability to share worries and fears about pain and the resulting physical disabilities can negatively influence the quality of life of a single, widowed or divorced person. Being accompanied or supported by a spouse would lead couples to be happier [39,40]. These results are consistent with the notion that an individual’s perceived quality of life may be good even if they have comorbidities or disabling symptoms. We did not find it in multivariable analysis but this can be justified by the recruitment of the population which is much younger; the patients included were >60 years old.

The World Health Organization also stresses the importance of social relations for older people; maintaining social relationships is a prerequisite for healthy aging, bringing life satisfaction. Specific studies of these characteristics in the Guadeloupian population could help refine our knowledge of the determinants of health and HRQoL in this population [41]. 

Our results provide an opportunity to emphasize the relevance of a systematic assessment of pain during home and office visits, whether by the general practitioner or the nurses.

Since pain negatively influences quality of life, optimal management of the former is essential, especially since older people report higher scores than younger adults. Given the data in the literature and our results, analogue scales are efficient tools, easily implemented in routine care. Indeed, the evolution of the scores is easy to trace in the medical record.

Nevertheless, a regular reassessment of the relevance of this tool is essential in the case of neurocognitive disorders. Moreover, managing pain in older adults to improve QoL is an objective to which GPs will be sensitive.

Our analysis presents several limitations. First, the design of our study does not allow for the determination of causal relationships between pain, IADL impairment and HRQoL. Second, self-reported health is a unidimensional proxy for HRQoL. As such, it fails to capture the multidimensionality of the latter. Nevertheless, despite a small number of participants (*n* = 115), study participants were representative of the older population in terms of pathologies and other associated comorbidities.

In regard to IADL dependency, the GPs’ interest remains rather modest in spite of many geriatricians, neurologists and others’ efforts for several years. Nevertheless, our result concerns the future of management strategies for IADL dependency because preserving QoL is fundamental for GPs.

## 5. Conclusions

This study was conducted on 115 patients aged 65 years and over, allowing the collection of demographic data regarding frailty syndrome, medication storage, cognitive function, algal complaints, functional status and health-related quality of life. Initial data found that complaints of pain and dependency to IADL were associated with a lower health-related quality of life. Although we found no association between suspected neurocognitive disorders and HRQoL, nor did we find an interaction effect between complaints of pain and neurocognitive disorders, detecting these three syndromes in community-dwelling older people is crucial for the prevention of adverse health events. Further studies are needed to confirm and consolidate our results.

## Figures and Tables

**Table 1 ijerph-20-03004-t001:** Baseline characteristics of study participants—KASADS.

Characteristics	Total (*n* = 115)
Age (years)	76.0 (±7.8)
Men	37 (32.2%)
BMI (Kg/m^2^)	26.8 (±5.3)
No diploma	39 (33.9%)
Lives alone	67 (58.3%)
Diabetes	50 (43.5%)
HTA	100 (87.0%)
Dyslipidemia	52 (45.2%)
HRQoL/100	66.2 (±20.3)
Pain/100	51.6 (±21.7)
IADL/4	3.4 (±1.0)
ADL	5.79 (±0.83)

*Notes:* Results are presented as means ± SDs or *n* (%). BMI: Body Mass Index; HRQoL: Health-related Quality of Life; IADL: Instrumental Activities of Daily Living; SD: standard deviation; HTA: Hypertension.

**Table 2 ijerph-20-03004-t002:** Associations between HRQoL score and patient’s characteristics.

Qualitative Variables	HRQoL (*n* = 115)	
Characteristics	Mean ± SD	*p*
**Sex**		
Men	68.6 ± 21.7	0.395
Women	65.1 ± 19.7	
**Marital status**		
Single	63.6 ± 21.4	0.103
Not single	70.0 ± 18.2	
**Educational level**		
No diploma	59.8 ± 23.1	0.014
≥1 diploma	69.6 ± 18.0	
**Age-related diseases**		
Diabetes		
Yes	68.4 ± 20.5	0.317
No	64.6 ± 20.2	
**Hypertension**		
Yes	66.3 ± 20.4	0.917
No	65.7 ± 20.3	
**Dyslipidemia**		
Yes	62.7 ± 19.6	0.086
No	69.2 ± 20.6	
**MMSE**		
Cognitive impairment (MMSE < 24)	64.7 ± 24.8	0.515
No cognitive impairment (MMSE ≥ 24)	66.9 ± 17.4	
**Quantitative variables**		
Characteristics	Correlation coefficient [CI]	*p*
Age	−0.092 [−0.272–0.093]	0.328
BMI	0.065 [−0.120–0.246]	0.49
Dependency to IADL	0.254 [0.074–0.419]	0.006
Pain	0.298 [0.121–0.457]	0.001

*Notes:* CI: 95% Confidence Interval; MMSE: Mini-Mental State Examination; HRQoL: Health-related Quality of Life; BMI: Body Mass Index; IADL: Instrumental Activities of Daily Living.

**Table 3 ijerph-20-03004-t003:** Determinants of HRQoL: multivariable analysis (linear regression model).

Explanatory Variable	Estimate (SD)	*p*
Dyslipidemia	−5.98 (3.56)	0.107
Pain	0.30 (0.08)	<0.001
No diploma	−2.14 (4.01)	0.606
Lives alone	−5.13 (3.64)	0.1174
Dependency to IADL	4.11 (1.81)	0.030

*Notes:* HRQoL: Health-related Quality of Life; IADL: Instrumental Activities of Daily Living. The estimates are the beta regression coefficients. When the beta is positive, it means that the variable concerned increases with the HRQoL score; when it is negative, it means that the variable evolves inversely to the HRQoL score.

## Data Availability

The datasets used and/or analyzed during the current study are available from the corresponding author upon reasonable request.

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
