# Peer review of "Correlates of Health-Related Quality of Life in Community-Dwelling Older Adults in Guadeloupe (French West Indies): Results from the KASADS Study"

_ijerph, 2023, doi:10.3390/ijerph20043004_

Round 1
Reviewer 1 Report
1. The aim of this study was to determine the correlates of Health-Related Quality of Life in community-dwelling older adults in Guadeloupe.
2. The increasing process of demographic ageing of the population brings with it new tasks for state and local administrations, which will require the formulation of objectives for such a policy that is conducive to meeting the specific needs of older people. A set of these needs characterises the quality of life of older people, which in everyday life is reflected in their health. Such an idea is certainly a necessary and relevant one.
3. The study was conducted in 115 patients aged 65 years and over, allowing the collection of demographic data regarding frailty syndrome, medication storage, cognitive function, algal complaints, functional status, and health-related quality of life. Initial data found that complaints of pain were associated with lower Health-Related Quality of Life.
4. The purposes stated by the Authors were confirmed in the conclusions. At the same time, the Discussion section has a discussion not only of the conclusions regarding the Authors' own research, but also of the research of other scholars, which would have been appropriate to move to the Literature Review section.
The conclusions are overly concise.
5. The literature cited in the work corresponds to the current research status in the field of the discussed issues.
Note that reference to source #15 is after reference to source #17, which breaks the overall sequence.
6. The literature review on the topic of the study, presented in the Introduction section, is rather superficial. It is based on only 11 sources, which is not sufficient for a publication of this level. Extending the literature review would automatically increase the length of the article, which in the presented version is a matter of discussion.
As mentioned earlier, the Discussion section should distinguish the discussion of the authors' own research from the achievements presented by other scholars in their publications. A "Literature Review" section should also be given emphasis.
Alternative analytical methods could be used for more in-depth conclusions, which could increase the attractiveness of the publication.
Author Response
Correlates of Health-Related Quality of Life in Community-Dwelling Older adults in Guadeloupe (French West Indies): Results from the KASADS Study
Dear Editor,
Please find appended a detailed point-by-point reply to the Referees and Editorial Board.
We are submitting a new version of our manuscript and we hope that the modifications meet your expectations.
Yours sincerely,
Maturin TABUE TEGUO on behalf of all authors.
Reviewer 1
Comments and Suggestions for Authors
- The aim of this study was to determine the correlates of Health-Related Quality of Life in community-dwelling older adults in Guadeloupe.
Author's response: We agree with the reviewer's suggestions, we make the changes in the text
- The increasing process of demographic ageing of the population brings with it new tasks for state and local administrations, which will require the formulation of objectives for such a policy that is conducive to meeting the specific needs of older people. A set of these needs characterises the quality of life of older people, which in everyday life is reflected in their health. Such an idea is certainly a necessary and relevant one.
Agree. We have modified the introduction section in accordance with the reviewer's comment
The study was conducted in 115 patients aged 65 years and over, allowing the collection of demographic data regarding frailty syndrome, medication storage, cognitive function, algal complaints, functional status, and health-related quality of life. Initial data found that complaints of pain were associated with lower Health-Related Quality of Life.
Agree. We make the changes considering the reviewer's remark.
- The purposes stated by the Authors were confirmed in the conclusions. At the same time, the Discussion section has a discussion not only of the conclusions regarding the Authors' own research, but also of the research of other scholars, which would have been appropriate to move to the Literature Review section.
Agree. We have modified the discussion section in accordance with reviewer's recommendations.
The conclusions are overly concise.
Agree. See the manuscript
- The literature cited in the work corresponds to the current research status in the field of the discussed issues.
Note that reference to source #15 is after reference to source #17, which breaks the overall sequence.
Agree. See the manuscript
- The literature review on the topic of the study, presented in the Introduction section, is rather superficial. It is based on only 11 sources, which is not sufficient for a publication of this level. Extending the literature review would automatically increase the length of the article, which in the presented version is a matter of discussion.
As mentioned earlier, the Discussion section should distinguish the discussion of the authors' own research from the achievements presented by other scholars in their publications. A "Literature Review" section should also be given emphasis.
Alternative analytical methods could be used for more in-depth conclusions, which could increase the attractiveness of the publication
In the recommendations to the author, it is not planned to put a chapter of the literature review. This review is already integrated into the discussion “Some variables that were not significant in multivariate analysis were not commented on, such as lack of a diploma and living alone. The inability to share worries and fears about pain and the resulting physical disabilities can negatively influence the quality of life of a single, widowed or divorced person. Being accompanied or supported by a spouse would lead couples to be happier. Hajian-Tilaki K, et al, Alan W. et al). These results are consistent with the notion that an individual's perceived quality of life may be good even if they have comorbidities or disabling symptoms. We did not find it in multivariate analysis but this can be justified by the recruitment of the population which is much younger, the patients included were >60 years old”.

Reviewer 2 Report
Initially, I think this is a very valuable and interesting study. It focuses on Community-Dwelling Older Adults in Guadeloupe (a region in the Caribbean), and determine the correlates of quality of life, so as to provide theoretical basis for improving quality of life of this population. However, after reviewing this manuscript, I found that some concerns needed to be further improved. The concerns are as follows:
1. Line 3 the word "adults" should be ”Adults”.
2. The ", " or "; " in lines 5-6 should be consistent.
3. Lines 24-25, 66,2 and 20,3 should be 66.2 and 20.3.
4. In introduction, The authors need to further explain why these independent variables of quality of life in lines 95-103 were selected, and whether these independent variables were selected based on theoretical models or their hypotheses.
5. In methods section, each variable in Lines 95-103 should be described in detail.
6. Line 107 "multivariate regression models" should be multivariable...", Other similar contents should be modified accordingly.
7. Why the authors include 115 older people? They should better supplement the sample size calculation method.
8. In Results section,The variables in table 1 should be presented completely,such as women, the score of ADL, etc.
9. In table 3 section, regarding "interaction effect between cognitive impairment and pain on quality of life", the authors should better present the results and describe the methods they used.
10. The authors should better describe the method used for correlation analyses, pearson or spearman?
11. Does the sample data conform to normal distribution? Why use parameter tests?
12. The discussion section should focus on the results and compared with the research results in other populations or races to highlight the particularity of the study samples,and the novelty of the study.
13. The section of Clinical impact of the study should be concise.
Author Response
Correlates of Health-Related Quality of Life in Community-Dwelling Older adults in Guadeloupe (French West Indies): Results from the KASADS Study
Dear Editor,
Please find appended a detailed point-by-point reply to the Referees and Editorial Board.
We are submitting a new version of our manuscript and we hope that the modifications meet your expectations.
Yours sincerely,
Maturin TABUE TEGUO on behalf of all authors.
Reviewer 2
Comments and Suggestions for Authors
Initially, I think this is a very valuable and interesting study. It focuses on Community-Dwelling Older Adults in Guadeloupe (a region in the Caribbean), and determine the correlates of quality of life, so as to provide theoretical basis for improving quality of life of this population. However, after reviewing this manuscript, I found that some concerns needed to be further improved. The concerns are as follows:
- Line 3 the word "adults" should be ”Adults”.
Author's response: We agree with the reviewer's comment, we make the changes in the text
- The ", " or "; " in lines 5-6 should be consistent.
Author's response: We agree with the reviewer's comment, we make the changes in the text
- Lines 24-25, 66,2 and 20,3 should be 66.2 and 20.3.
Author's response: We agree with the reviewer's comment, we make the changes in the text
- In introduction, The authors need to further explain why these independent variables of quality of life in lines 95-103 were selected, and whether these independent variables were selected based on theoretical models or their hypotheses.
Author's response: As with pain, studying the assessment of quality of life in populations with cognitive impairment can be challenging due to the disorders presented by the subjects. A person's quality of life is determined by factors that are both objective (standard of living, state of health, functional abilities, participation and social contacts, leisure activities, etc.) and subjective factors (satisfaction with health, meaning of life).
- In methods section, each variable in Lines 95-103 should be described in detail.
Author’s response: Agree with the reviewer’s comment. We have described the IADL scale, Katz’ADL scale and MMSE scale: “We used the Instrumental Activities of Daily Living scale (IADL)[12] and Katz’ ADL scale [13] to assess the functional status of the participants. and the Mini-Mental. The Lawton’s IADL scale measures 8 items: using the telephone, shopping, preparing food, housekeeping, doing laundry, using transportation, handling medications and handling finances. The score is from 0 to 8, a score of 8 indicates total autonomy, 0 indicates total dependency. The Katz’ ADL scale is a 6-item scale assessing independence in 6 basic ADL: bathing, toileting, transferring, eating, dressing, incontinence. For each activity, a score of 1 indicates complete autonomy, a score of 0.5 indicates partial autonomy, and a score of 0 indicates complete dependency. The Mini-Mental. State Examination (MMSE) to assess cognitive functions. This 30-item scale assesses the severity of cognitive deterio-ration through items of orientation, learning, attention and arithmetic, memory, language, and constructive praxis. The scale is rated from 0 to 30, reflecting the level of cognitive impairment.”
- Line 107 "multivariate regression models" should be multivariable...", Other similar contents should be modified accordingly.
Author’s response: We agree with reviewer’s comment. We have developed statistical analysis part:
“Quantitative variables were expressed as mean ± standard deviation, and qualitative variables were expressed as percentage. Student’s t-test was used to assess the significance of associations between categorical and continuous variables, and Pearson’s correlation was used to assess the correlation between continuous variables. Variables at a 20%- threshold in the univariate analysis were considered in a multivariable analysis using a linear regression. At last, an interaction term between pain and cognitive impairment on quality of life was searched in a multivariable analysis using a linear regression model. Missing data were not imputed. We used a fixed p-value cut-off of 0.05 to determine significance. All analyses were performed with the RStudio software (v.3.0.2. 21).”
- Why the authors include 115 older people? They should better supplement the sample size calculation method.
Author’s response: Sample size calculation was based on the main study assessing the association between drug storage and frailty. For the purpose of this study, we have used these data in order to assess the correlates of quality of life. We have added the reference of the main study (in press) which details the methodology of the study, notably sample size.
- In Results section,The variables in table 1 should be presented completely,such as women, the score of ADL, etc.
Author’s response: according to the reviewer’s comment, we have added the ADL score. In order not to overload the table, we have only put male sex, diabetes=yes etc. If the reviewer and the editor consider that it hinders the reading of the table, we will complete the table.
- In table 3 section, regarding "interaction effect between cognitive impairment and pain on quality of life", the authors should better present the results and describe the methods they used.
Author’s response: We had initially hypothesized that an interaction was possible between cognition and pain on quality of life based on our experience and literature (Rostad et al. Associations between Pain and Quality of Life in Severe Dementia: A Norwegian Cross-Sectional Study. Dement Geriatr Cogn Dis Extra. 2017 Apr 7;7(1):109-121). We therefore tested this interaction in a linear regression model. The interaction term was not significant (p=0,741). Method and results have been modified accordingly.
- The authors should better describe the method used for correlation analyses, pearson or spearman?
Author’s response: Fully agree with reviewer’s comment. The Pearson’s test was used since at least one of the two variables followed a normal distribution in each association.
- Does the sample data conform to normal distribution?Why use parameter tests?
Author’s response: Yes, at least one of the two variables followed a normal distribution in each association. In addition, we have at least 30 patients for each of the variables. It is why we used parametric tests. Using non-parametric tests do not change our results on p-value in terms of significance.
- The discussion section should focus on the results and compared with the research results in other populations or races to highlight the particularity of the study samples,and the noveltyof the study.
Author's answer: Yes, in the discussion we focus first on our results obtained which we then compare with the results of research done in other populations.
Some variables that were not significant in multivariable analysis were not commented on, such as lack of a diploma and living alone. The inability to share worries and fears about pain and the resulting physical disabilities can negatively influence the quality of life of a single, widowed or divorced person. Being accompanied or supported by a spouse would lead couples to be happier ( Hajian-Tilaki K, et al, Alan W. et al). These results are consistent with the notion that an individual's perceived quality of life may be good even if they have comorbidities or disabling symptoms. We did not find it in multivariate analysis but this can be justified by the recruitment of the population which is much younger, the patients included were >60 years old.
- The section of Clinical impact of the study should be concise
Author's response: Indeed, the paragraph on clinical impact seems long because limitations of the study was included in this section; We therefore removed the title Clinical Impact from the study, and move the limitation section just above; that lengthened the paragraph

Round 2
Reviewer 1 Report
The limitations of the research that the Authors introduced in the second version of the article are to be commended. The empirical part of the article has been expanded and strengthened. However, some comments have been partially corrected, namely:
1) the Discussion section has a discussion not only of the conclusions regarding the Authors' own research, but also of the research of other scholars, which would have been appropriate to move to the Literature Review section.
2) The literature review on the topic of the study, presented in the Introduction section, is rather superficial. It is based on only 11 sources, which is not sufficient for a publication of this level.
3) The length of the article is a matter of discussion.
Author Response
The limitations of the research that the Authors introduced in the second version of the article are to be commended. The empirical part of the article has been expanded and strengthened. However, some comments have been partially corrected, namely:
1) the Discussion section has a discussion not only of the conclusions regarding the Authors' own research, but also of the research of other scholars, which would have been appropriate to move to the Literature Review section.
The literature review on the topic of the study, presented in the Introduction section, is rather superficial. It is based on only 11 sources, which is not sufficient for a publication of this level.
Author's response: We agree with the reviewer's suggestions. The discussion has been reinforced by commentaries, which makes the discussion part more complete. We have also reinforced the introduction section by adding references which provide more clarifications on HR-QoL concept (1. Ko Y, Lee K. Int J Environ Res Public Health. 2022 May 6;19(9):56-59; 2. Ko H, Jung S, Int J Environ Res Public Health. 2021 Jan 19;18(2):818; 3.
15;12(12):e0189648)
2) The length of the article is a matter of discussion
Author's response: We agree with the reviewer's suggestions. Adding comments actually increased the length of the discussion.
Reviewer 2 Report
First of all, I would like to thank the authors for making a lot of amendments to their manuscript and for responding to my comments one by one.
This is a topic worthy of study. I hope the authors will carry out more in-depth research on the population in this area in the future, such as cancer population, chronic diseases population, etc.
After reviewing the authors' revised manuscript, I still have many problems that need to be further solved by the authors.
#1 Line 3,adults should be “Adults” ; The letter "a" should be capitalized.
#2 Line 5, Gebhard Pierre 24 should be 2,4
#3 Line 6, et Tabue-Teguo Maturin, what means "et"?
#4 Lines 45, 59-61, The authors mentioned that pain was related to health-related quality of life, so what is the significance of the author's current study? I suggest that the authors should study the relationship between different types, different degrees and different body parts of pain and quality of life in the future.
#5 Line 102-105, The contents in the manuscript and cover letter are inconsistent, please check!The total score is 4 or 8 ?
#6 Line 134, Table 1,the score of ADL is 5.79 or 5.10? please check!
#7 Table 3, what statistics does "Estimate" represent? Standardized coefficient or others? Please explain the meaning of these data in the discussion section.
#8 Can linear regression model do interaction effect? How to do?
#9 Line 175-176, the authors mentioned that " Although this association is often found in the literature, alleviating pain and com-plaints of pain does not necessarily improve quality of life.", Why should the authors study pain instead of selecting other variables?
#10 The conclusion should be concise.
#11 Please check the format of reference 36.
#12 Further improve Table 1 and Table 3.
I have a minor suggestion. I hope the authors will pay attention to the format of the manuscript, punctuation marks, etc.
Best wishes & good luck
Author Response
First of all, I would like to thank the authors for making a lot of amendments to their manuscript and for responding to my comments one by one.
This is a topic worthy of study. I hope the authors will carry out more in-depth research on the population in this area in the future, such as cancer population, chronic diseases population, etc.
After reviewing the authors' revised manuscript, I still have many problems that need to be further solved by the authors.
#1 Line 3,adults should be “Adults” ; The letter "a" should be capitalized.
Author's response: We agree with the reviewer's suggestions, we make the changes in the text
#2 Line 5, Gebhard Pierre 24 should be 2,4
Author's response: We agree with the reviewer's suggestions, we make the changes in the text
#3 Line 6, et Tabue-Teguo Maturin, what means "et"?
Author's response: We agree with the reviewer's suggestions, we move “et” in the text
#4 Lines 45, 59-61, The authors mentioned that pain was related to health-related quality of life, so what is the significance of the author's current study? I suggest that the authors should study the relationship between different types, different degrees and different body parts of pain and quality of life in the future.
In this work, we are mainly interested in the determinants of quality of life which is our variable of interest. Again, thank you for the reviewer suggestions who will help us to continue our work in this population. (Boucaud-Maitre D, Cesari M, Tabue-Teguo M, Lancet Heathly Longev 2023 ; Boucaud-Maitre D et al, JMIR Res Protocol, 2023 ; Vainqueur L, Simo N et al. Front Med 2022)
According to the reviewer’s comment, we added this sentence in the discussion section (limitation of our study) “It would have been interesting to study the relationship between HRQoL and the different aspects of pain (type, intensity, location, etc.”)
#5 Line 102-105, The contents in the manuscript and cover letter are inconsistent, please check!The total score is 4 or 8 ?
Thanks, the total score of IADL is 4.
#6 Line 134, Table 1,the score of ADL is 5.79 or 5.10? please check!
Thanks, the score is 5.79 (+/- 0.83)
#7 Table 3, what statistics does "Estimate" represent? Standardized coefficient or others? Please explain the meaning of these data in the discussion section.
Author's response: The estimates are the beta regression coefficients. When beta is positive, it means that the variable concerned increases with the HRQoL score; when it is negative, it means that the variable evolves inversely to the HRQoL score. We have added in the footnote of table 3
#8 Can linear regression model do interaction effect? How to do?
Author's response: Yes, linear regression is used to test interaction effects. However, we did not have an interaction hypothesis to test a priori.
#9 Line 175-176, the authors mentioned that " Although this association is often found in the literature, alleviating pain and com-plaints of pain does not necessarily improve quality of life.", Why should the authors study pain instead of selecting other variables?
The idea for us was to say that quality of life is multidimensional and that our work is intended to open new perspectives. We tested other variables, but pain was more significantly associated with univariate HRQoL. Only variables with a certain univariate significance level were included in the multivariate model.
#10 The conclusion should be concise.
#11 Please check the format of reference 36.
Author's response: We agree with the reviewer's suggestions, we make the changes
#12 Further improve Table 1 and Table 3.
I have a minor suggestion. I hope the authors will pay attention to the format of the manuscript, punctuation marks, etc.
Thanks. We make all changes in the manuscript